# Effect of Sliding on the Relation of Tribofilm Thickness and Wear

Martin Jech [1,*], Maria L. Miranda-Medina [1], Thomas Wopelka [1], Christian Tomastik [1] and Carsten Gachot [2]

1    AC2T Research GmbH, Viktor-Kaplan-Strasse 2, 2700 Wiener Neustadt, Austria
2    Institute of Engineering Design and Product Development, Lehargasse 6, 1060 Vienna, Austria
*    Correspondence: martin.jech@ac2t.at

**Abstract:** The formation of tribofilms depends on temperature, shear stress, availability of the related chemical components, and characteristics of the near surface region, e.g., roughness and surface chemistry. The purpose of a tribofilm is to separate two sliding surfaces, thus preventing or limiting wear. This research article aims for the first time at a systematic approach to elucidate on a fundamental level the interplay between tribofilm formation in particular thickness and wear behavior in the boundary and mixed lubrication regime. For this, load, temperature and sliding frequency as most relevant parameters are taken into consideration. For that purpose, a piston ring and cylinder liner configuration in an oscillating tribometer was chosen as a model system, with the top dead centre conditions in internal combustion engines of passenger cars as the testing regime. The amount of wear produced during the tribotests is continuously monitored by means of the Radio-Isotope Concentration (RIC) method. The tribofilm is investigated via Atomic Force Microscopy (AFM), Scanning Electron Microscopy with Energy Dispersive X-ray Spectroscopy (SEM-EDS) and X-ray Photoelectron Spectroscopy (XPS). The results clearly indicate that the impact of load on the wear rate can be seen in an Archard-like dependency, but changes of temperature and sliding velocity in the boundary to mixed lubrication regime imply a non-linear ratio between wear and tribofilm formation.

**Keywords:** piston ring–cylinder liner; Radio-Isotope Concentration method; Archard's wear model; tribofilm formation ZDDP; tribotesting

## 1. Introduction

Zinc dialkyl dithiophosphates (ZDDP) are the most widely used anti-wear additives for combustion engine oils [1–4]. After several decades of use, emerging detrimental effects of ZDDP and an increased necessity for ZDDPs have been in a precarious balance in recent years. On the one hand, the tribofilms formed from ZDDP are known to increase friction losses in the engine [5,6]. On the other hand, efforts to reduce friction losses have led to an increased use of ultra-low-viscosity engine oils. This in turn results in higher stress on engine surfaces, therefore requiring the still ongoing use of ZDDP anti-wear additives. As a further consideration, the exhaust products originating from ZDDPs are undesirable both because of their poisoning of exhaust catalysts and for environmental reasons [4,7]. Therefore, investigation of ZDDPs is still of high importance. Thanks to extensive research, much is already known about the processes involved in the anti-wear function of ZDDP, in particular regarding the structure of the ZDDP-derived tribofilms [8–13]. Broadly speaking, these tribofilms consist of glassy phosphates that form on the tribologically loaded surfaces from fragments of the ZDDP molecules, which serve as precursors for this purpose. It is known that temperature as well as shear stress play key roles in the formation of the tribofilms [8,14–17].

The properties of ZDDPs and ZDDP-derived films have been investigated via different approaches, e.g., taking chemical hardness [18], temperature [14,16,19], sliding [20], test

duration and cycles [15,21], or contact load [14,22] into account. Tribofilm thickness has been determined, e.g., by XPS analysis [10,23], spectroscopic ellipsometry [24] and other optical methods [5,8,9,13,16,24,25]. The range of wear measurement methods in this context ranges from simple strategies, e.g., evaluating the weight loss of the contact bodies after the tribological interaction [26], and optical methods [27], to more sophisticated techniques where the wear can be monitored on-line by using radioactive isotopes [28–32].

Recent research still deals with many different aspects of the topic. Gosvami et al. [14] showed that by increasing the number of sliding cycles, the volume of the tribofilm increases linearly, while an increase in contact pressure and temperature has an exponential effect on the growth rate of the tribofilm monitored via AFM. Ueda et al. [9] investigated the various states of crystallinity of ZDDP-derived films using TEM and correlated these with chemical, mechanical and friction as well as wear properties of the films. Zhang et al. [12] measured the boundary friction of ZDDP-derived films with particular regard to the effect different alkyl chains have on the frictional properties of the films. Hsu et al. [23] re-investigated the chemical structure of ZDDP-derived films employing atom probe tomography, allowing them to confirm and refine the knowledge about the typical layer structure of the films.

Experimental investigations are supplemented by numerous model calculations and numerical simulations on tribofilm growth [17,33,34]. Mohammadtabar et al. [22] presented molecular dynamics simulations explaining the individual contributions of heat, load, and shear force in a sliding interface under the effect of an extreme-pressure additive. The results indicated that all three factors accelerate the chemical reaction of the additive; however, under moderate conditions the tribofilm formation is governed more by mechanical forces than by frictional heating. Pu et al. [16] developed a model for the generation, flow and transfer of frictional heat in a piston ring–cylinder liner system with ZDDP film growth. In this study, the authors showed that by taking local flash temperature and asperity contact into account, a significantly improved model for tribofilm growth can be developed. Chen et al. [11] calculated tribofilm formation and removal on a cylinder liner surface based mainly on mechanical properties and surface roughness to investigate the wear protection the film provides depending on the system parameters.

While wear in terms of removal of tribofilm has been extensively investigated so far in research articles, wear of the actual initial sliding surface, once covered by the tribofilm after a running-in phase, has not been regarded by itself and is the subject of this research work. Moreover, while the overall wear of the substrate may still be determined post-experiment, the evolution of substrate wear during a tribological test is usually not accessible. In particular, the evolution of the wear of the substrate below the tribofilm is a quantity that is by now in some cases considered in model simulations, but hardly accessible for experimental measurements. For example, Azam et al. [34] include the time-resolved substrate wear in their extensive plastoelastohydrodynamic lubrication model in the form of a modified Archard equation and arrive at a distinct running in and steady-state behaviour that has yet to be verified experimentally.

In this contribution, a systematic study of the wear behaviour of the system piston ring against cylinder liner with further attention on tribofilm formation under various loading conditions (load, temperature, sliding frequency) is presented for the first time to connect the interplay between tribofilm formation and particularly tribofilm thickness and wear in the boundary and mixed lubrication regime depending on the aforementioned parameters. For this purpose, the focus is mainly on the cylinder liners, as they show higher wear than the piston rings in this configuration. The variation of loading conditions is performed through a fractional factorial study by means of a specialised tribometer providing oscillating movement. Wear is monitored continuously via a radioactive isotope concentration method. Wear behaviour is considered on the one hand in terms of the wear model of Archard [35], according to which harsher loading conditions (such as higher load and higher temperature) correlate with higher wear. On the other hand, increasing temperature and increasing load (indicating shear stress) are thought to lead to

the formation of thicker tribofilms [14,22], which should in turn reduce wear. This study tries to improve the understanding of this balance of tribofilm-affected wear behaviour.

## 2. Materials and Methods

### 2.1. Tribological Testing

Tribological tests were carried out with an SRV4® (Optimol instruments) reciprocating sliding tribometer, with a custom-built piston ring (PR) and cylinder liner (CL) sample holder (see Figure 1b). The piston ring samples were mounted in the upper sample holder, which was driven in a linear oscillatory motion, while the cylinder liner samples were fixed in the lower sample holder. The holders are within a soft polymer chamber, ensuring that there is no oil leakage during the wear measurement. The engine oil was supplied to the contact via an enclosed lubricant circuit to maintain a continuous flow of the lubricant during the tribotests (see Figure 1a).

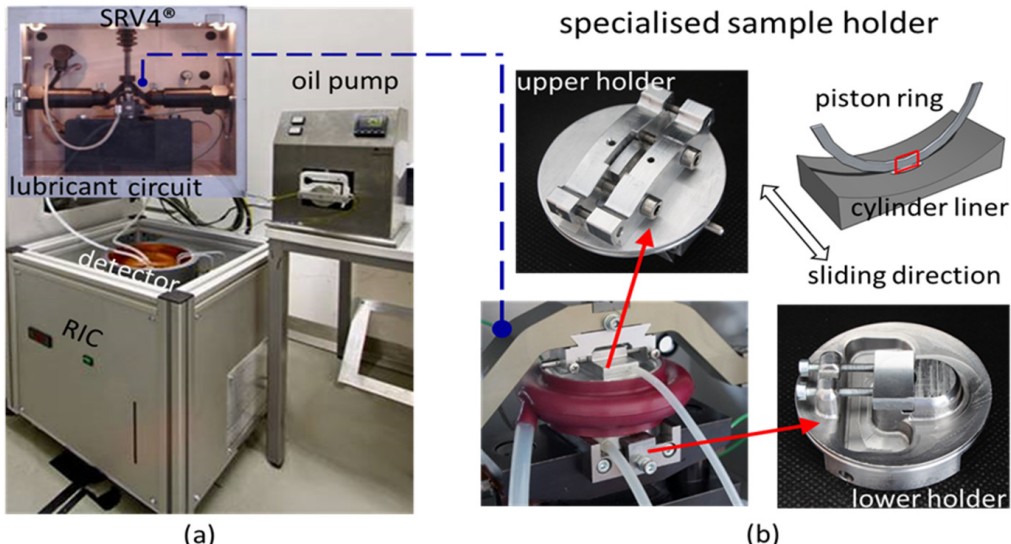

**Figure 1.** Experimental set-up for tests performed on (**a**) a reciprocating sliding tribometer with a (**b**) specialised sample holder to preserve a constant flow of the lubricant in the contact zone between the piston ring and cylinder liner samples.

The presented research focusses on the pressure regimes and the wear rates occurring in the interaction between the piston ring and cylinder liner at ignition and top dead centre (TDC) conditions in the engine [32]. Considering the material combination as well as the geometry involved in the contact zone between the cylinder liner and piston ring, the average stress was calculated by using the software Hertzwin [36]. For a normal load of 50 N, the average stress resulted in 90 MPa, while for ~200 N it is equivalent to ~310 MPa. However, in the tribometer set-up the influence of the combustion, e.g., in the form of chemical products and related follow-up effects, or the influence of the engine stroke, leading to scratches of loose wear particles in the hydrodynamic region, or long-term effects like fatigue or embrittlement are not considered.

The experiments in this paper focus on distinct conditions, ranging from 20 °C (room temperature), 6 Hz oscillating frequency and 50 N load up to 120 °C, 28 Hz and 200 N. Piston rings of nitrided steel (X90CrMoV18, with a hardness HV 0.05 of 1200 at the surface (nitrided) and 410 in the bulk 150 μm below the surface, and a Ra roughness value of $0.12 \pm 0.01$ μm in sliding direction) and cylinder liners of cast iron (with a hardness HV10 of $235 \pm 11$, and a Ra value of $0.19 \pm 0.04$ μm in sliding direction) were selected for these tests. The study was planned as a fractional factorial study (see Table 1), giving all the tests a comparable total sliding distance. Originally, the high frequency tests had been planned at 25 Hz (representing 1500 ignition points per minute corresponding to 3000 rpm),

but because of observed resonance effects of the tribometer test set-up the frequency was shifted to 28 Hz. By keeping the original total test time, a resulting relative change in the total sliding distance of 12% was obtained.

**Table 1.** Tribometer test parameters.

| Oil Temperature (°C) | Load (N) | Frequency (Hz) | Test Duration (h) | Stroke (mm) | Sliding Distance (m) |
|---|---|---|---|---|---|
| 20 | 50 | 6 | 25 | 3 | 3240 |
| 20 | 200 | 6 | 25 | 3 | 3240 |
| 20 | 200 | 28 | 6 | 3 | 3628 |
| 120 | 50 | 28 | 6 | 3 | 3628 |
| 120 | 200 | 28 | 6 | 3 | 3628 |
| 120 | 200 | 6 | 25 | 3 | 3240 |

For each test, fresh samples of piston ring and cylinder liner have been taken. Before the tests, all samples were cleaned with petroleum ether in an ultrasonic bath for 30 min. The lubricant used in all tests was fully formulated mineral engine oil, SAE grade 5W30 (middle SAPS concentration (sulphated ash, phosphorus and sulphur), no Mo anti-wear additives. For each test, an amount of approximately 60 mL was used in the lubricant circuit. The physical-chemical properties and the elemental composition of the lubricant are shown in Table 2.

**Table 2.** Basic physical-chemical properties of the fully formulated engine oil.

| Physical Properties | |
|---|---|
| Kinematic viscosity (mm$^2$/s) | 40 °C: 56.2 |
| | 100 °C: 9.84 |
| TAN, (mg KOH/g) | 3.6 |
| TBN, (mg KOH/g) | 10.5 |
| Antiwear additive | primary ZDDP |
| Friction modifier | no Mo and B containing compounds |
| Base oil group | III |
| Chemical Elements (ppm) | |
| Zn | 1190 |
| Ca | 3860 |
| Mg | <10 |
| B | <10 |
| P | 1065 |
| Mo | 0 |
| Cl | <30 |

*2.2. Wear Measurements*

Wear of the piston ring and cylinder liner samples were measured continuously during the tests by means of the Radio Isotope Concentration method (RIC, AC2T research GmbH) [37]. This technique is based on activating an area, which contains the wear zone, with a known concentration of radioactive isotopes and a known depth distribution (depth profile) of the radioactive isotopes. The RIC device is attached to the tribometer set-up (see Figure 1a). Wear particles, which contain the radioactive isotopes, are transported to a gamma detector via the closed lubricant circuit. The radioactive isotope used in this study was Co57, as this can be produced from iron at the sample surface by thin layer activation (TLA) [31]. The detector measures the activity of the wear particles in the lubricant and subsequently the wear depth or wear volume is calculated on the basis of the radioisotope depth profile. The wear depth is shown as a graph over time, in which a running-in and a steady-state wear phase are distinguished (see Figure 2). In the steady-state wear regime,

a quasi-static condition can be assumed, for which the tribological conditions do not change and thus a steady wear rate can be observed. Consequently, wear follows a linear trend in the steady-state regime as a function of time and is fitted with linear least squares fitting. The steady-state wear rate is defined as the slope of the fitted curve in the steady-state wear phase and as a best practice approach, the slope is calculated over the last 40% of the registered data, where steady-state behaviour is assumed. The running-in wear is defined as the difference of the extrapolated steady-state trend and zero wear at the starting point of the tribological testing (see Figure 2). The integral of the steady-state wear rate over testing time is the steady-state wear.

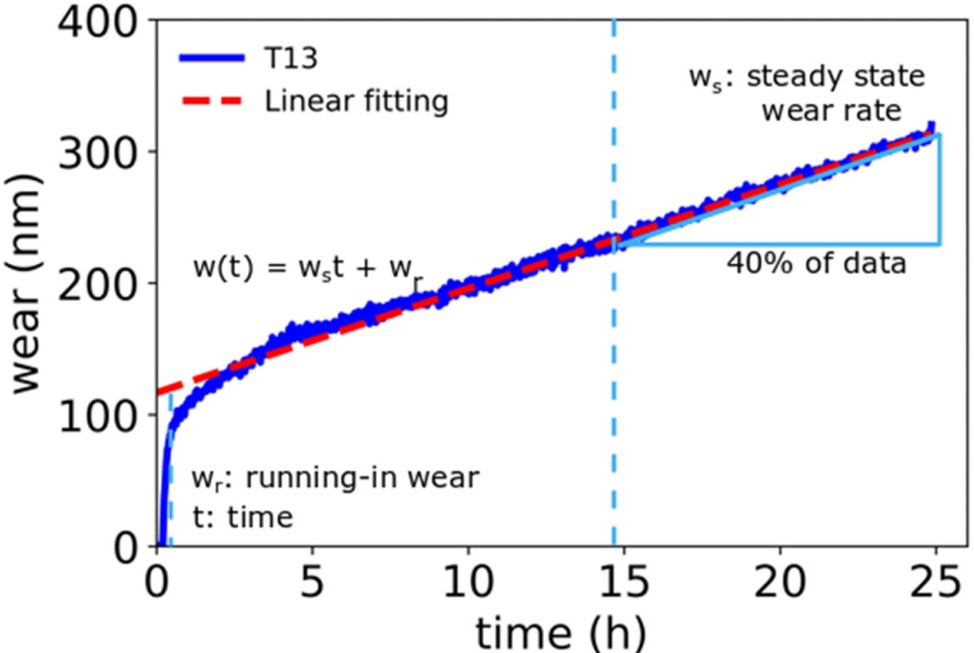

**Figure 2.** Definition of the running-in and steady-state wear obtained from a wear graph measured via the Radio Isotope Concentration method (RIC).

After the tests, all samples were rinsed in petroleum ether to remove the residual oil. Thus, the tribofilm analysed via surface analysis is the chemisorbed tribofilm; the physisorbed tribofilm is not accessible with this cleaning and analysis procedure.

*2.3. Surface Analysis*

After the tribometer tests, selected samples were probed in air with an atomic force microscope (AFM, Veeco, CP-II, New York, NY, USA) with a silicon tip (spring constant: 0.9 N/m, f0 = 13–27 kHz) in the contact mode, with a scan rate of 1 Hz and a load of 100 nN. Sample areas of $100 \times 100 \ \mu m^2$, $50 \times 50 \ \mu m^2$, $20 \times 20 \ \mu m^2$ and $10 \times 10 \ \mu m^2$ were scanned close to the centre of the wear scar, as well as out of the wear scar.

Additionally, scanning electron microscopy (SEM, JEOL JSM-T330A, Tokyo, Japan) and energy dispersive X-ray spectroscopy (EDS, Oxford Instruments, Oxford, UK) were performed on selected cylinder liner samples, with an electron beam voltage of 10 kV, to analyse the tribofilms on the worn areas as well as the spatial distribution of the additive-derived elements in the tribofilms.

Furthermore, two piston ring samples and their respective cylinder liner counterparts, representing two extremes of the loading conditions, were chosen for X-ray photoelectron spectroscopy (XPS) analysis. XPS data were acquired using a Thermo Fisher Scientific Thetaprobe with a monochromatic Al Kα X-ray source (1486.6 eV). High-resolution spectra were obtained at 50 eV pass energy with an energy step size of 0.2 eV. Peak fitting was performed with the Thermo Fisher Scientific Avantage Data System software, us-

ing Gaussian/Lorentzian curve fitting for the evaluation. The C1s peak for adventitious carbon at 284.6 eV was used as a binding energy reference. Peak backgrounds were subtracted using a modified Shirley algorithm [38]. Sputter depth profiles were acquired using 3 keV Ar ions.

The set of measurements and analyses performed on the cylinder liner and piston ring for each group of tribotest parameters are summarised in Table 3, outlining in total 17 independent tests.

**Table 3.** Summary of tribometer tests and surface analysis performed on the piston ring (PR) and cylinder liner (CL) sample. The notation T# originates from the numbering of the performed tests and was also used for analysis sample designation. Tests with activated cylinder liner and activated piston ring were carried out in separate tests.

| Tribotest Parameters | 20 °C-6 Hz | | | | 20 °C-28 Hz | 120 °C-28 Hz | | | | 120 °C-6 Hz |
|---|---|---|---|---|---|---|---|---|---|---|
| | 50 N | | 200 N | | 200 N | 50 N | | 200 N | | 200 N |
| Activated for RIC | CL | PR | CL | PR | CL | CL | PR | CL | PR | CL |
| Tribometer test | T7 T10 | T1 | T8 T13 | T3 | T14 T18 T23 | T5 T9 | T4 | T6 T11 | T2 | T15 T16 |
| SEM | ✓ | - | ✓ | - | - | ✓ | - | ✓ | - | - |
| AFM | ✓ | ✓ | ✓ | ✓ | - | ✓ | ✓ | ✓ | ✓ | - |
| XPS | ✓ | ✓ | - | - | - | - | - | ✓ | ✓ | - |

## 3. Results

### 3.1. Effect of Load, Frequency, and Temperature on Friction and Wear Behaviour

3.1.1. Load

The results of the friction coefficient over time obtained from tests with similar frequency and temperature are shown in Figures 3 and 4. After the stabilisation due to running-in, the friction coefficients tend to be constant for all conditions during the whole experiment. The difference in the loads of 50 N and 200 N does not significantly influence the value of the friction coefficients. Thus, the effect of load is not significantly detectable in the coefficients of friction. Superficially, a coefficient of friction is expected to be independent of the load for conditions similar to the conditions applied here. As such, this result is not surprising and the friction results in this paper are documented for the sake of completeness, but the focus is laid on the more sensitive wear measurements.

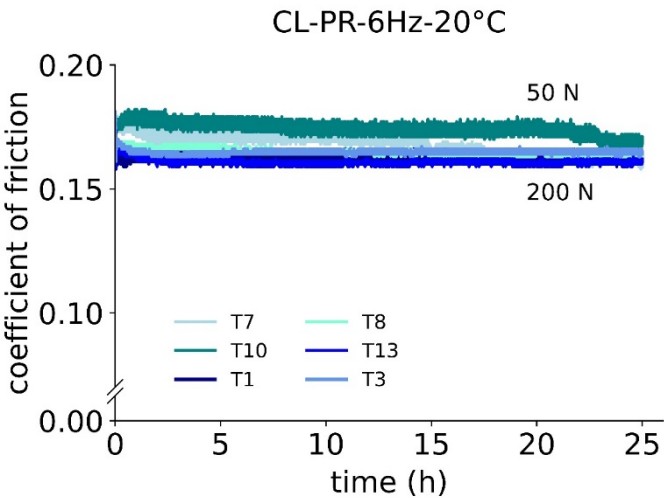

**Figure 3.** Coefficients of friction at 6 Hz and 20 °C with respect to loads of 50 N and 200 N.

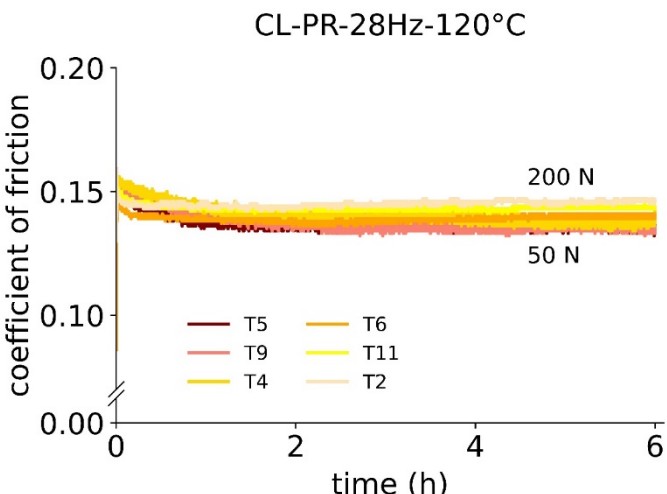

**Figure 4.** Coefficients of friction at 28 Hz and 20 °C with respect to loads of 50 N and 200 N.

Standardised coefficient of friction values (COF) obtained via an SRV4 tribometer are derived from the highest values of the measured friction force (roughly top 10%) and averaged over several strokes [39]. Consequently, the obtained COF values are related mainly to the relatively high friction forces at the turning points of a stroke, where low relative velocities implicate a reduction of the lubricating film. The COF values are related to a lesser degree to the moderate friction forces at the mid-stroke region of a stroke, where hydrodynamic effects become more important. Thus, the values of coefficient of friction are mostly determined by solid-solid contact and such by boundary lubrication conditions, rather than by mixed or hydrodynamic conditions. Summarising, the standardised COF values are higher, as they should be for such a tribosystem.

In Figures 5 and 6, the wear graphs of the cylinder liner samples are shown as separate sets for the same respective frequency and temperature each, emphasising the effect of load. It is obvious that a higher load leads to higher wear, especially regarding running-in wear but also recognisable for the steady-state wear rate. These wear graphs also show the high repeatability of the combination wear measurement and tribometer set-up, as the tests with the same test parameters are perfectly comparable to each other and clearly distinguishable from tests with different tests parameters.

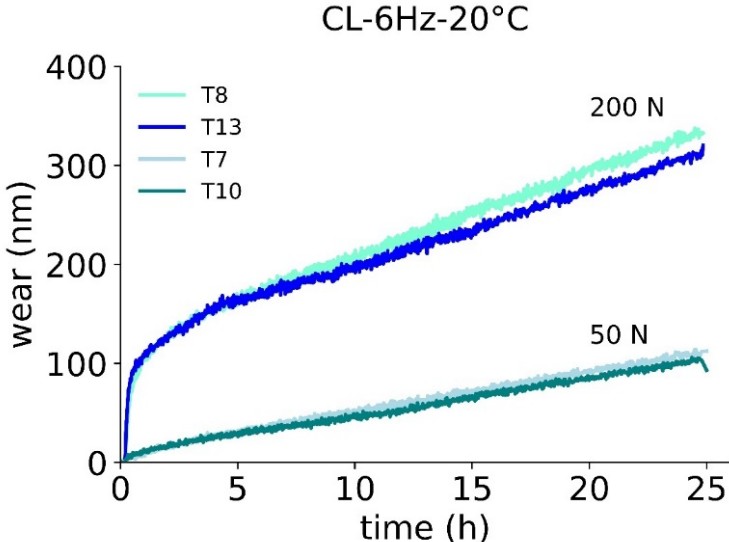

**Figure 5.** Wear of cylinder liner samples at 6 Hz and 20 °C with respect to loads of 50 N and 200 N.

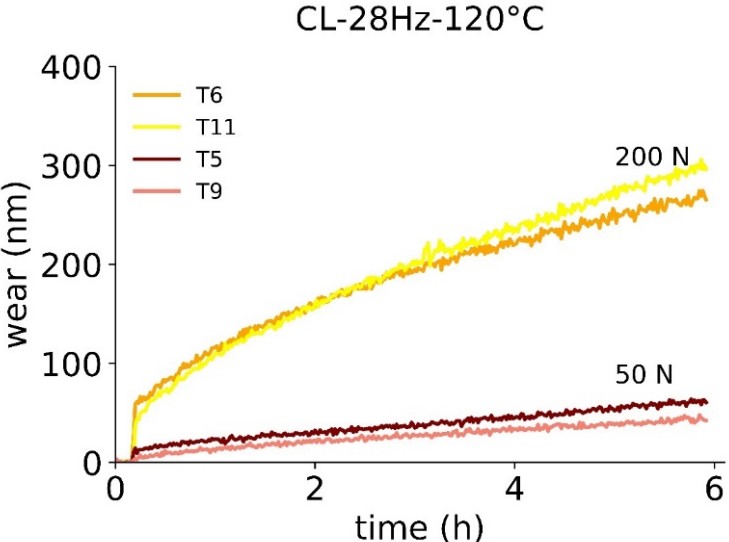

**Figure 6.** Wear of cylinder liner samples at 28 Hz and 120 °C with respect to loads of 50 N and 200 N.

The wear graphs of the piston rings, Figures 7 and 8, show the same load dependency as observed for the cylinder liners. The wear of the piston rings is one order of magnitude smaller than the wear of the cylinder liners. This is in accordance with our experience of investigating the piston ring and cylinder liner system in such tribometer environments and loading conditions. This is due to the hardness of the samples, the nitrided steel of the piston rings being harder than the grey cast iron of the cylinder liner pieces.

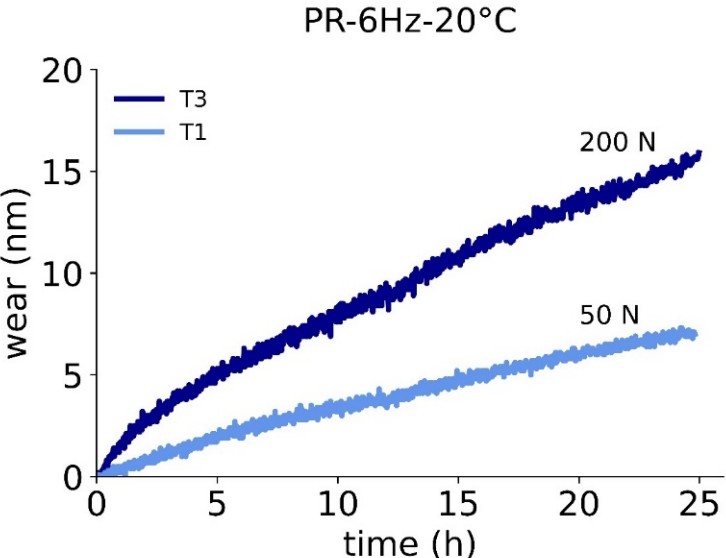

**Figure 7.** Wear of piston ring samples at 6 Hz and 20 °C with respect to loads of 50 N and 200 N.

The difference in wear due to load, Figures 5–8, at the same frequency and temperature, is in agreement with the load dependency described by Archard´s classical wear model [35]. It is given by the equation:

$$W_V = \mathrm{k} * \left( \frac{SL}{3H} \right) \tag{1}$$

where $W$ is the wear, $S$ is the sliding distance, $L$ is the load, k is a proportionality constant in (m$^3$/Nm) and $H$ is the hardness of the material. Since the wear parameter $W$ can be obtained experimentally, the derived variable in Equation (1) is the proportionality constant k, which is in the order of 4·10–16 ± 1·10–16 m$^3$/Nm for the steady-state wear of the

cylinder liners, taking H= 248 (N/m$^2$) for the cylinder liner from the literature [40], and on the order of 6.5·10–17 $\pm$ 1·10–17 m$^3$/Nm for the steady-state wear rate of the piston rings.

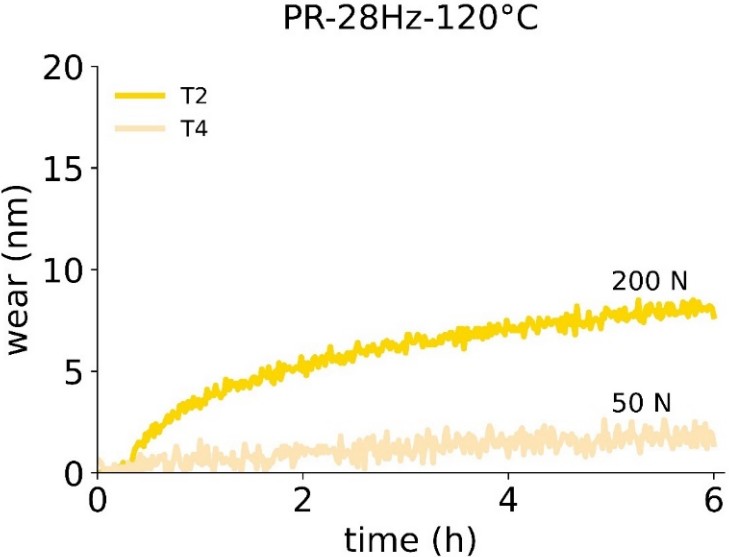

**Figure 8.** Wear of piston ring samples at 28 Hz and 120 °C with respect to loads of 50 N and 200 N.

3.1.2. Frequency

To emphasise the impact of the sliding frequencies, the x-axes are displayed as sliding distance in Section 3.1.2 Frequency. Regarding the 20 °C wear graphs (see Figure 9) an increase of the sliding frequency leads to a decrease of wear, both running-in wear and steady-state wear rate. This is in accordance with the expected Stribeck-like behaviour. A higher frequency corresponds to a higher sliding velocity. With higher velocity, the system gets closer to hydrodynamic lubrication or, in other words, the real fraction of solid–solid contacts of the mixed lubrication regime decreases. Consequently, friction and wear decrease in the mixed lubrication regime with higher velocity. This is in accordance with the trends of the friction coefficients for 20 °C (see Figure 10).

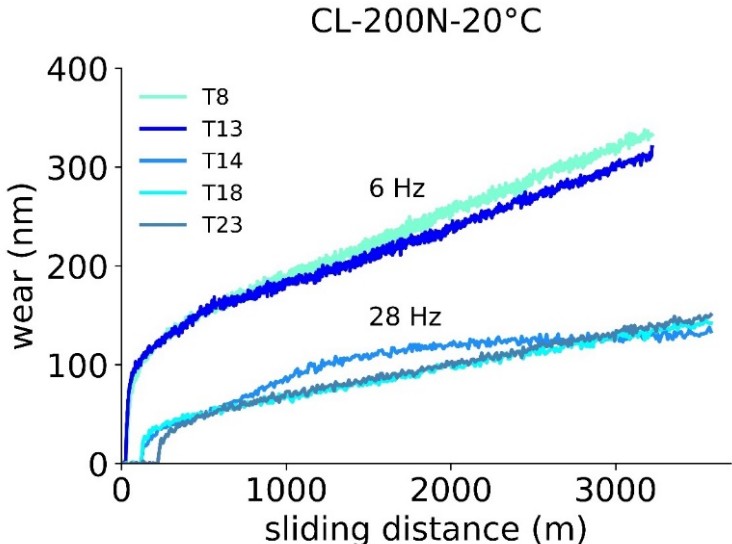

**Figure 9.** Wear of cylinder liner samples at 200 N and 20 °C with respect to frequencies of 6 Hz and 28 Hz.

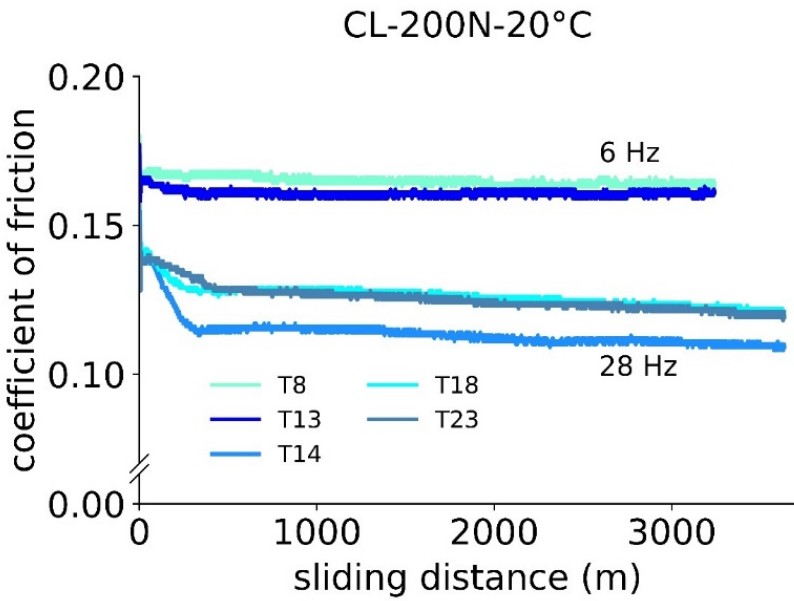

**Figure 10.** Coefficients of friction at 200 N and 20 °C with respect to frequencies of 6 Hz and 28 Hz.

For 120 °C, a decrease of wear due to increase of frequency is observable only for the steady-state wear rate (Figure 11), but less pronounced as for the 20 °C tests. The friction coefficients for 120 °C (see Figure 12) are in accordance with the expected Stribeck-like behaviour, but likewise less pronounced than for the 20 °C tests. For the tests at 120 °C, the same argument as for the 20 °C tests can be applied, that higher frequency leads to a higher hydrodynamic influence. For 120 °C, this effect seems to be less pronounced as for 20 °C, as the viscosity of the lubricant is lower, and the mixed lubrication regime consequently extended to higher sliding velocities.

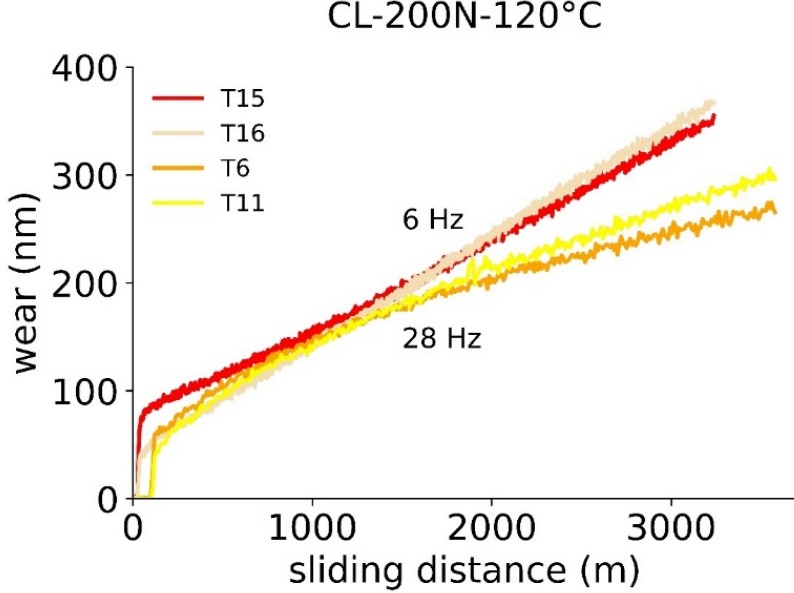

**Figure 11.** Wear of cylinder liner samples at 200 N and 120 °C with respect to frequencies of 6 Hz and 28 Hz.

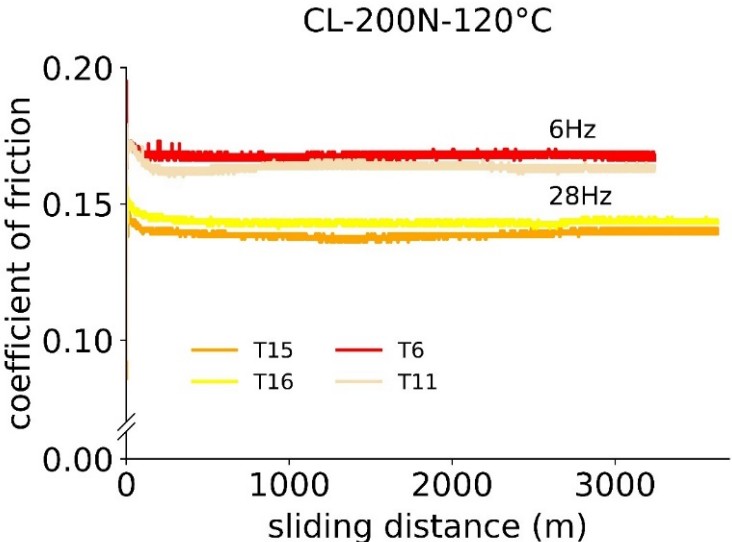

**Figure 12.** Coefficients of friction at 200 N and 120 °C with respect to frequencies of 6 Hz and 28 Hz.

3.1.3. Temperature

The influence of temperature on wear appears in various ways, as temperature affects the material properties, especially hardness, the viscosity of the lubricant, and the chemistry of the tribofilm formation. The influence of the temperature-related hardness change is neglected, as it is regarded to be of minor impact in the applied temperature range compared to the viscosity changes or tribofilm formation. As the tribofilm formation will be discussed later in this paper, the focus in this chapter is on the temperature-related viscosity effect.

Starting with the tests at 28 Hz, the increase of temperature leads to an increase of wear (see Figure 13) but also to an increase of friction (see Figure 14). The increase of temperature goes along with a decrease of viscosity of the engine oil (see Table 2) and thus a higher friction at the same sliding frequency, as expected for a Stribeck-like behaviour in the mixed lubrication regime. Consequently, the increase of wear can be understood as an increase of solid–solid contacts.

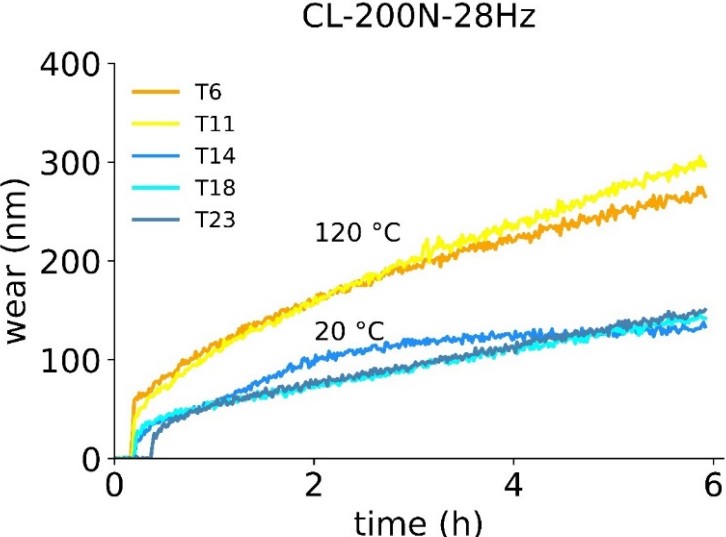

**Figure 13.** Wear of cylinder liner samples at 200 N and 28 Hz with respect to temperatures of 20 °C and 120 °C.

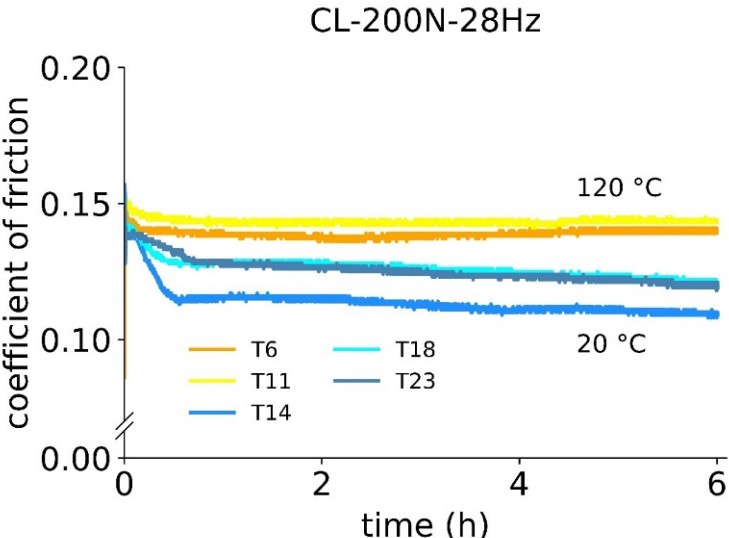

**Figure 14.** Coefficients of friction at 200 N and 28 Hz with respect to temperatures of 20 °C and 120 °C.

Regarding wear (see Figure 15) and friction (see Figure 16) for tests at 6 Hz, hardly any difference can be observed for the tests with different temperatures. Either the tests at 6 Hz are still in the boundary lubrication regime, so that a temperature-dependent viscosity does not have a significant impact, or the tests are marginally in the mixed lubrication regime and the temperature-related effects of viscosity and tribofilm formation compensate each other in terms of wear and friction. As the surfaces, in particular the roughness of the surfaces, change with running-in wear and tribofilm formation, the application of a single lambda ratio over the whole test duration for distinguishing the regimes of boundary and mixed lubrication is regarded to be insufficient here. Zhu and Wang reported on the lambda ratio under mixed lubrication conditions and emphasised that the previously taken simplifications lead to inaccurate values for the lambda ratio under mixed conditions. The lambda ratio in turn is calculated based upon average film thickness and composite roughness of the interacting sliding surfaces and in particular the roughness is continuously changing in the contact. Zhu and Wang therefore defined an upper limit to be in the range 0.6–1.0 and a lower limit 0.01–0.05 [41].

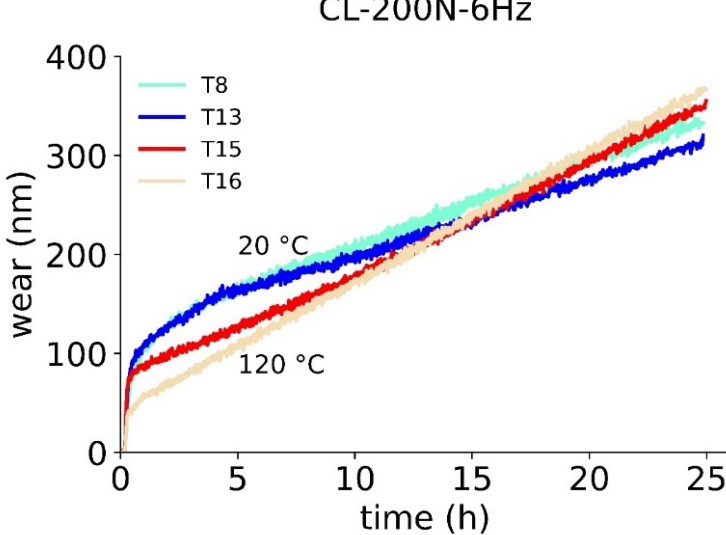

**Figure 15.** Wear of cylinder liner samples at 200 N and 6 Hz with respect to temperatures of 20 °C and 120 °C.

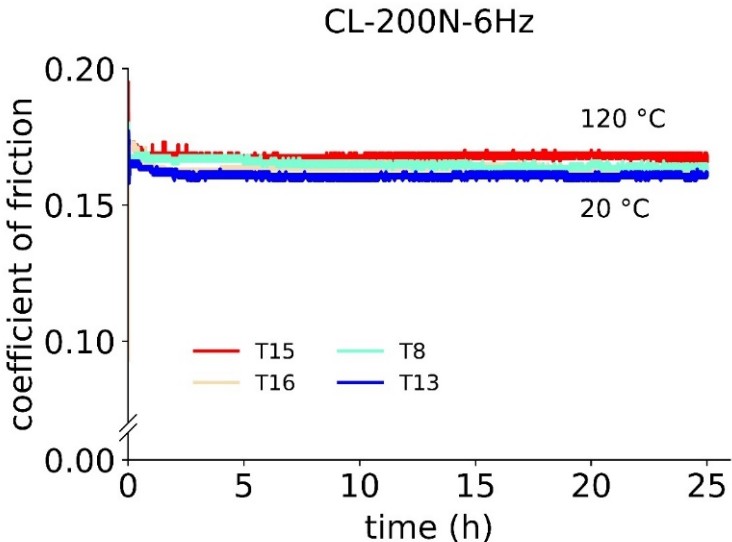

**Figure 16.** Coefficients of friction at 200 N and 6 Hz with respect to temperatures of 20 °C and 120 °C.

*3.2. Surface Analysis Performed on Cylinder Liners and Piston Ring Samples*

3.2.1. Analysis of the Surfaces by SEM

A detailed SEM analysis was conducted for the tribotest performed at 120 °C (see Figure 17). Elemental mappings acquired via EDS show that a section of the area contains the elements P, S, and Zn, which can be directly correlated to a ZDDP-induced tribofilm. In the tribofilm section sliding marks (scratches) can also be observed (vertical direction in the figure). Apart from the tribofilm, the initial surface with honing marks (grooves in diagonal direction in the left area of measured area) is also visible.

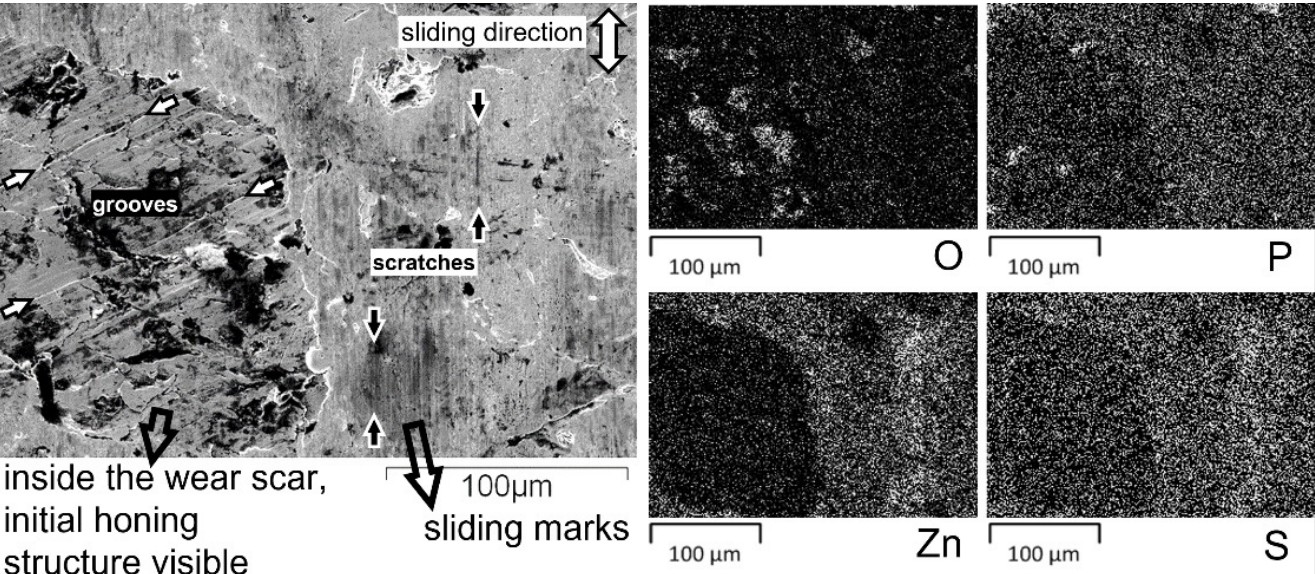

**Figure 17.** EDS elemental mappings of a cylinder liner surface after a tribometer test performed at 120 °C (**right** side). The SEM image of the area inside the wear scar (**left** side) reveals both initial honed surface (**left** part of the image) and an area covered by a tribofilm (**right** part of the image). Examples of grooves of the original honed surface and sliding scratches on the tribofilm are marked to guide the eye.

### 3.2.2. Surface Topography of the Samples by AFM

Figure 18 shows examples of surface topography images obtained via AFM. Honing marks (diagonal grooves) from the original surface can be recognised on the cylinder liner surfaces for tests carried out at 50 N (Figure 18, top, cylinder liners on left side). However, when the load is increased to 200 N (Figure 18, bottom) the honing marks are hardly visible anymore on the tribofilm-covered cylinder liner surfaces (CL). For the piston ring surfaces (PR) a patch-like structure is apparent (Figure 18a,b), images on the right side). The lateral size of the patches increases with load and temperature. By comparing the images of cylinder liners (left) and piston rings (right in Figure 18), the cylinder liner tribofilms are composed of more coalesced and irregularly shaped pads, whereas on the piston rings the pads appear more distinct and evenly dispersed on the surface of the wear scar.

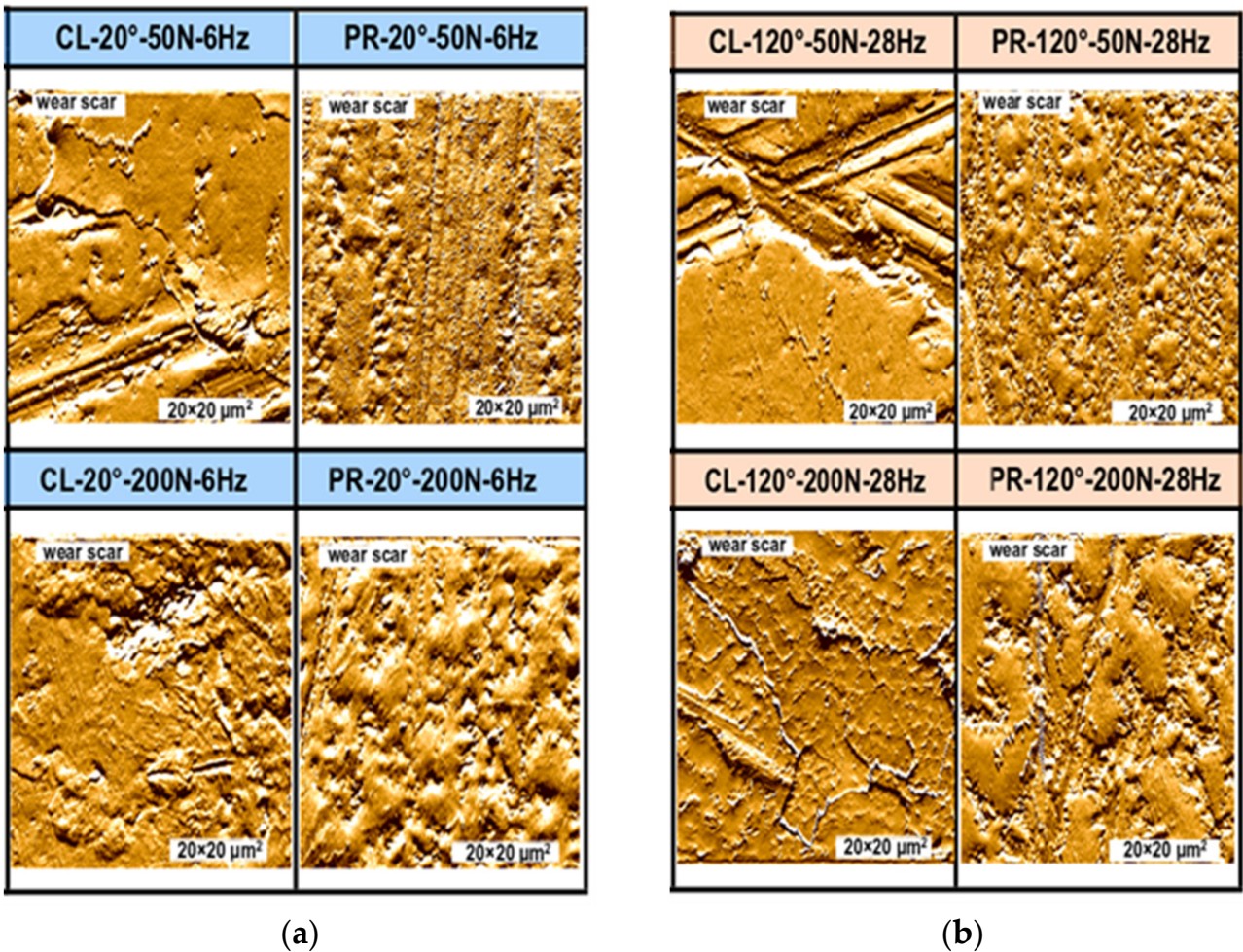

**Figure 18.** Atomic force microscope images obtained from the surface of cylinder liners (CLs) and piston rings (PRs). The samples are grouped by the temperature of the tests: (**a**) 20 °C (blue) and (**b**) 120 °C (orange). Sliding direction is vertical.

### 3.2.3. Comparison of Tribofilm Structure Obtained with AFM and SEM

In Table 4 an overview of the SEM analyses is given, comparing inside and outside of the wear scar and regarding different loading conditions.

The tribofilm inside the wear scar, where shear is provided by the load and sliding contact, is more apparent than for the film outside the wear scar, which forms only due to temperature. Inside the wear scar, the tribofilm partially covers the honing marks and also shows marks in the sliding direction.

When comparing the AFM pictures with high resolution (Figure 18) with the SEM and AFM pictures of moderate resolution (Table 4) the tribofilms seem to have different features on (at least) two lateral length scales; first, a patch-like "micro" structure in the range of 1 to 10 μm, and second, a coalescent "meso" structure larger than 100 μm (see also Figure 17).

The rows in (Table 4) are sorted by the obvious presence of the tribofilm inside the wear scar, from low to high coverage. This sorting correlates to the loading conditions increasing from mild to severe, when taking load as the main parameter and temperature as secondary parameter into account.

**Table 4.** AFM and SEM analysis of cylinder liner surfaces—comparison of different loading conditions and surfaces inside the wear scar and outside the contact zone (The pictures are orientated in sliding direction going vertical and honing marks going diagonal.).

| | SEM | | AFM | |
|---|---|---|---|---|
| | out of the contact zone | wear scar | wear scar | out of the contact zone |
| 50 N 6 Hz 20 °C | $260 \times 180\ \mu m^2$ | $260 \times 180\ \mu m^2$ | $100 \times 100\ \mu m^2$ | $100 \times 100\ \mu m^2$ |
| 50 N 28 Hz 120 °C | $260 \times 180\ \mu m^2$ | $260 \times 180\ \mu m^2$ | $100 \times 100\ \mu m^2$ | $100 \times 100\ \mu m^2$ |
| 200 N 6 Hz 20 °C | $260 \times 180\ \mu m^2$ | $260 \times 180\ \mu m^2$ | $100 \times 100\ \mu m^2$ | $100 \times 100\ \mu m^2$ |
| 200 N 28 Hz 120 °C | $260 \times 180\ \mu m^2$ | $260 \times 180\ \mu m^2$ | $100 \times 100\ \mu m^2$ | $100 \times 100\ \mu m^2$ |

### 3.2.4. Tribofilm Analysis by XPS Depth Profiling

Two representative cases of the most extreme tribological loading conditions were investigated by means of XPS depth profiling, as due to the tribological conditions the most discriminative tribofilm properties are assumed. As such, a tribofilm representing conditions of low temperature, low frequency and low load (Figure 19 top) is compared to a tribofilm representing conditions of high temperature, high frequency and high load (Figure 19 bottom).

To obtain a measure for the thickness of the tribofilms from the sputter depth profiles, an Fe concentration of 50 at% (the main element in the cast iron bulk of the cylinder liners or the steel bulk of the piston rings, respectively) was chosen as the point, where the depth value was taken for that purpose, as indicated in Figure 19. It can be observed that the

thickness of the tribofilm is thicker for the higher loading condition at the piston ring as well at the cylinder liner, as expected.

By means of XPS, the elemental binding states are also available for a detailed analysis, as shown as an example for sulphur in Figure 20. Although sulphur is present to a similar amount at the cylinder liner and the piston ring tribofilms, the sulphate states at 168 eV have only been recognised at the piston rings. Without further proof, this may be due to the fact, that the piston ring surface is always in contact during the oscillating movement and as such suffering a continuous impact of shear compared to an equivalent area of the cylinder liner surface. This shear impact may also be the reason for the different surface morphology of the tribofilms observed on the piston ring and cylinder liner surfaces (see Figure 18) and different tribofilm thickness (see Figure 19).

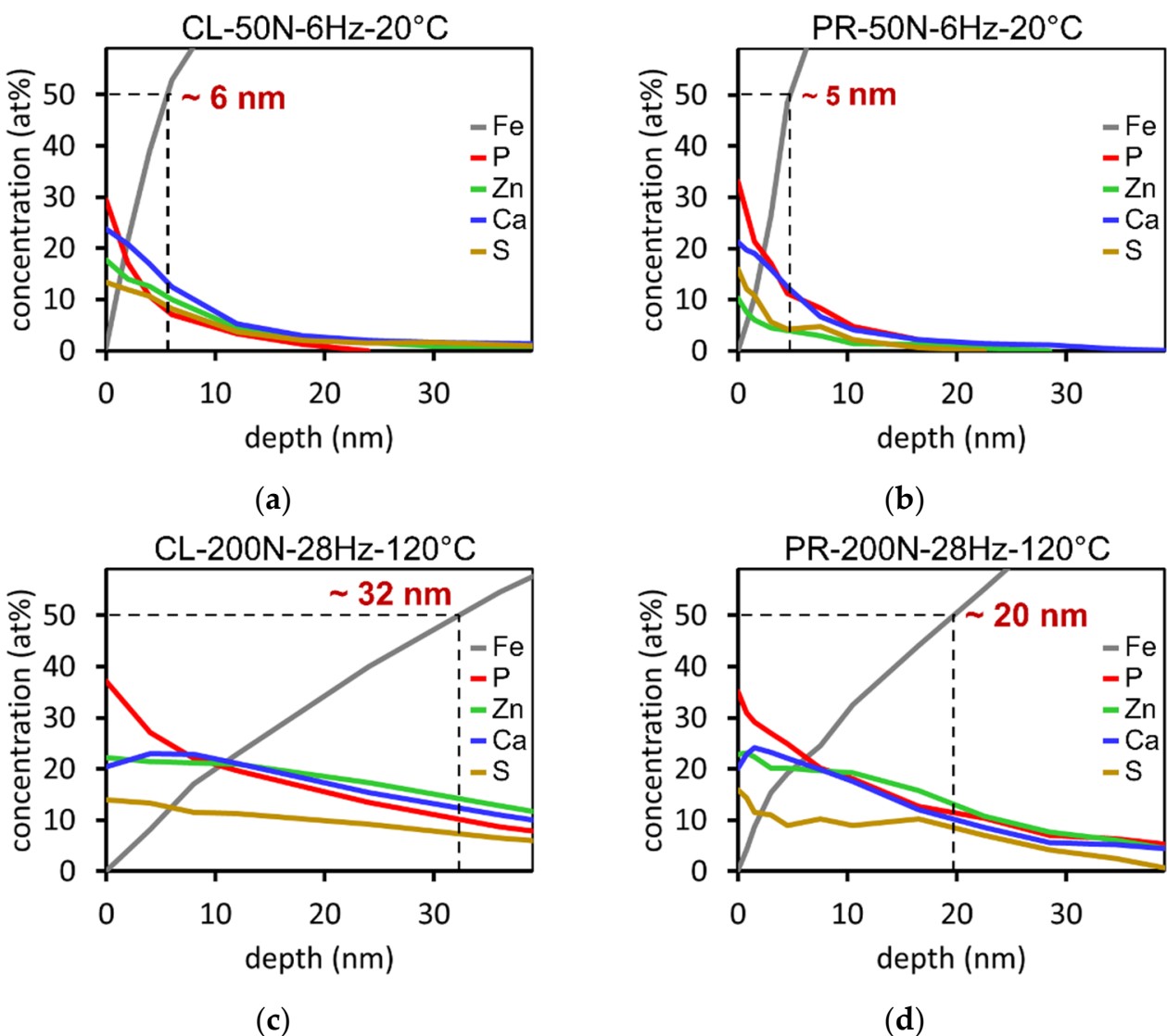

**Figure 19.** X-ray spectroscopy depth profiles of cylinder liner (**a**–**c**) and piston ring (**b**,**d**) surfaces. Tribofilm thickness estimated at 50% of Fe.

For the sake of completeness, it is noted here that the sulphate state can only be detected in the first few nanometers of the tribofilm and is below the significance level in the "bulk" of the tribofilms.

When comparing the XPS findings (3.2.4) with SEM (3.2.1) and AFM (3.2.2), it has to be noted that the XPS analysis represents an average over a measuring spot of roughly 400 μm

in diameter. Therefore, the use of XPS does not allow for an unambiguous distinction between smaller local inhomogeneities in composition or tribofilm coverage.

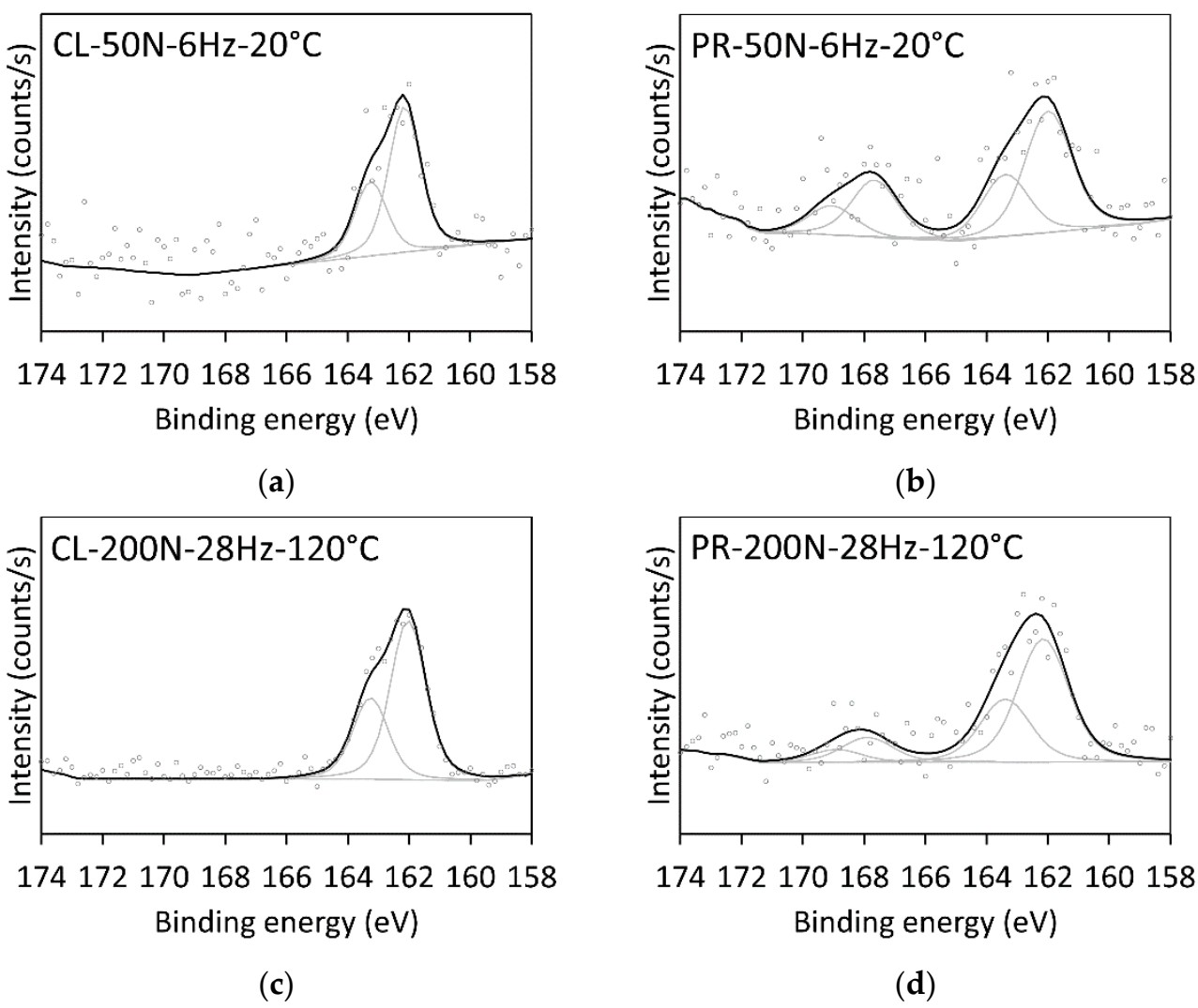

**Figure 20.** X-ray photoelectron spectra of S acquired during depth profiling of cylinder liner (**a**–**c**) and piston ring (**b**,**d**) surfaces, taken at a depth of approx. 4 nm, fitted with 2p3/2-2p1/2 peak doublets; circles: measured data; grey lines: fitted component peaks; black lines: fitted sum envelope; peaks at 162 eV originate from sulphide states, peaks at 168 eV from sulphate states.

## 4. Discussion

### 4.1. Friction

In summary, regarding the friction results, (1) load hardly has an impact on the COF, (2) higher frequency leads to lower COF, and (3) higher temperature leads to higher COF at least at 200 N and 28 Hz. These trends can be understood regarding a Stribeck-like behaviour and boundary to mixed lubrication conditions.

However, it should be noted that the running-in periods are hard to estimate simply from the COF curves, as most of the COF curves hardly show any change with time. As such, the obtained COF curves are not used for correlating the tribometric data with tribofilm formation but are still useful in terms of reproducibility of the performed experiments.

### 4.2. Wear

In Figure 21, the steady-state wear rate in nanometers per hour is presented in ascending order. Looking at the x-axis and the order of the loading conditions, load is the determining parameter, followed by the temperature and the sliding frequency (relative velocity). With more severe loading conditions, which means higher load, higher temperature and higher sliding frequency, a higher wear rate is observed. As a first conclusion, this is the expected wear behaviour and is proven as well for this study.

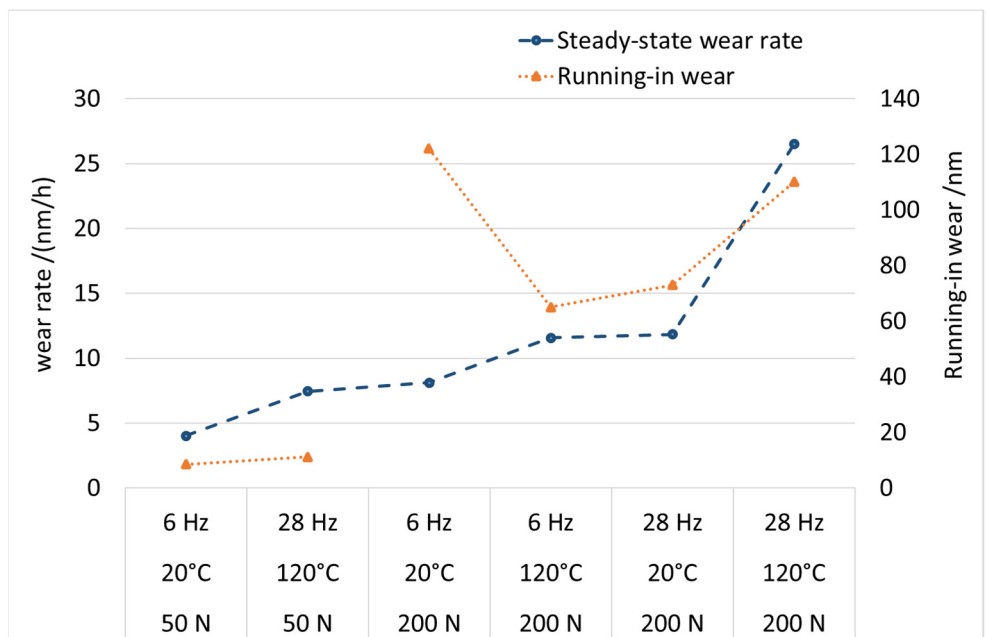

**Figure 21.** Steady-state wear rate and running-in wear over loading conditions; loading conditions sorted by increasing wear rate; the data points are connected by a dashed line for easier visibility.

The running-in wear is displayed in Figure 21 as well and is separated into two branches for 50 N and 200 N load. The running-in wear of the test with 20 °C, 200, and 6 Hz (third from left) seems to disrupt a continuous trend and thus a correlation with the wear rate. As there are two decently congruent tests (T8 and T13) confirming this measurement, the running-in value is not regarded as an outlier.

### 4.3. Tribofilm

The appearance of the tribofilms (see Table 4) in terms of thickness and coverage is in accordance with the state-of-the-art knowledge that an increase of temperature and shear (load, frequency of sliding cycles) leads to a thicker tribofilm [14].

This knowledge does not help to understand the wear trend, as observed in Figure 21. Low temperature and low shear lead to a thin tribofilm, as observed for the test condition with 20 °C, 50 N and 6 Hz (Figure 19a), and as such should imply the lowest wear-preventing ability of the tribofilm. Conversely, high load conditions, where a thick tribofilm is observed (Figure 19c), should lead to high wear-preventing ability and therefore to low wear. Apparently, this is not the full truth as a thicker tribofilm does not automatically lead to low wear rates (Figure 21). This leads to the conclusion that wear and tribofilm thickness do not directly correlate with each other. According to Morina et al., the thicker tribofilm should lead to a higher COF but less wear [42]. As the tribofilm growth is mechanically or shear activated, the growth starts ceasing once the necessary stress as driving force is not sufficient anymore and plastic deformation sets in. Because of the growth and the increasing pad area, there might be also weakened bonds in the tribofilm, thus resulting in a reduced structural film integrity with therefore higher wear observed. One of the basic questions finally is if the tribofilm has to be fully intact or dense or even at a certain

thickness as it is more important to have a continuous film build up and replenishment of the contact with tribofilm material [43].

### 4.4. Tribofilm and Wear

With the applied RIC method, the loss of the initially activated material into the lubricant is measured. Thus, the evolution of the measured wear with test duration depends not only on immediate time-dependent effects of the contact, such as, e.g., wear of the asperity tips, geometric adaption, and/or tribomutation, but also on the developing ability of the tribofilm to protect the initial material from wear.

Regarding the applied tribological conditions, three possible mechanisms of wear of the initial surface can be identified, when a tribofilm is present:

1. The initial surface is not completely covered by the tribofilm and exposed to direct contacts with the counter-acting body. This is in accordance with the topographic images obtained with AFM and SEM (see Table 4). All the images show features and even blank areas of the initial surface besides the patch-like tribofilm.
2. The tribofilm is too thin or not strong enough to prevent asperities of the counteracting body to plough through it and scratch the initial surface.
3. The initial surface chemically reacts with the tribofilm. Rubbing off the tribofilm consequently leads to an increase of worn material of the initial surface in the lubricant, which is subsequently measured via the RIC method.

Within this work, none of these mechanisms of wear can be emphasised or neglected. However, the wear curves show a clear running-in behaviour. The wear process starts with a relatively high wear rate that decreases with time or sliding distance, respectively.

As soon as both friction and wear running-in phases are concluded, a steady-state regime can be observed, which in turn allows for estimating the running-in wear (see also Figure 2). Additionally, when a steady-state regime can be supposed, we may as well assume that the tribofilm likewise reaches a steady state, with stationary formation and removal rates balancing each other. Conversely, running-in behaviour is then related to tribofilm build-up.

For discussing the impact of running-in on wear, the wear coefficient based on the steady-state wear is compared to the ratio of running-in and total wear (see Figure 22). As all these tests have a similar sliding distance, the wear coefficient and the share of running-in the total wear should give an insight. There is no wear coefficient of the running-in wear, as running-in changes with time and as there is no clear point in time when running-in is finished.

Figure 22 reveals that there is not an obvious trend between wear coefficient and running-in share. Firstly, the diagram shows that the data separates into two groups, the first at 6 Hz and the second at 28 Hz sliding frequency. This implies that the sliding frequency in the form of the lubrication condition, rather than shear stress, is decisive regarding wear and tribofilm formation. This is a crucial point as the lubrication conditions change during tribological experiments due to the change in roughness. The change in roughness is further due to wear but also the forming tribofilm. Additionally, the tribofilm may have most likely different material properties compared to the initial surface, being rather close to the lubricant properties and thus contributing to the lubrication film.

Secondly, the wear coefficient is not the same for all the tests. This implies that the wear model of Archard would need an extension taking the tribofilm formation and lubrication condition into account. Such a new model cannot be set up on the basis of the results of this study, as the fractional factorial study only reveals the effect of the studied parameters. For setting up such a wear model, an extensive and thorough study needs to be carried out with multiple parameter variations (e.g., at least three to four values of each parameter) with a subsequent tribofilm analysis of all the specimens.

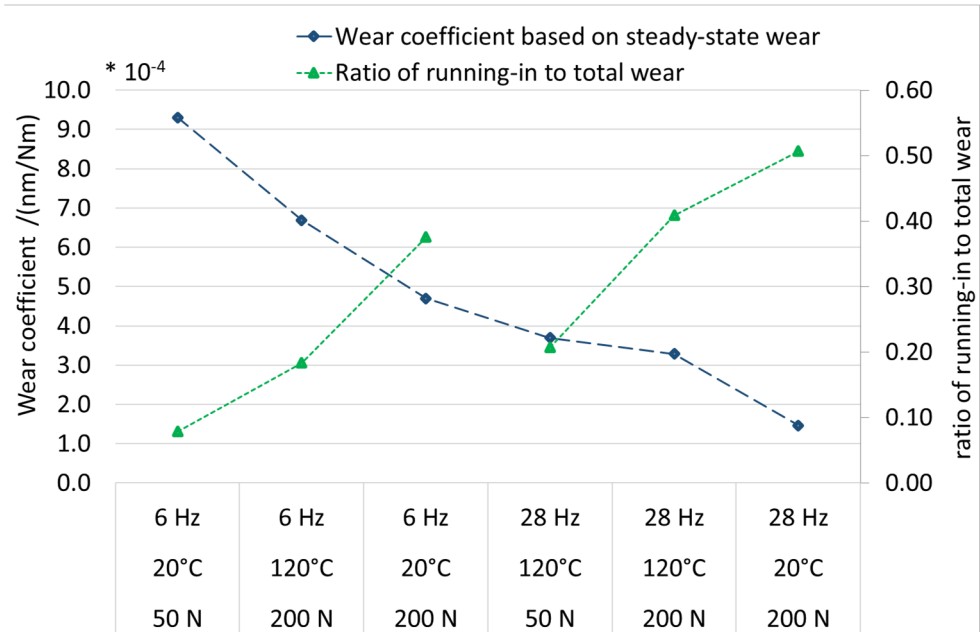

**Figure 22.** Wear coefficient in comparison to loading conditions and the ratio of running-in wear to total wear; the data points are connected by a dashed line for easier visibility.

Additionally, such a wear model would need to take the underlying mechanisms of wear into account, especially the removal of the substrate, formation of the tribofilm, removal of the tribofilm, as well as possible re-deposition of wear particles and tribofilm fragments onto the surface. Such processes either need an in situ surface analytical technique, capable of characterising the changes on the surface during the tribological contact, or a computer-based simulation, bridging the gap between known and accessible conditions of the tribosystem, such as the condition before and after the tribological loading. The presented research may not clarify the underlying friction and wear mechanisms, but definitely provides the necessary information, especially regarding the wear rates with respect to different loading conditions, so that computer-based models can be evaluated and enhanced.

## 5. Conclusions

A fractional factorial test series regarding load, temperature, and sliding frequency was carried out on an oscillating tribometer for the tribosystem piston ring against cylinder liner. The test parameters were chosen to simulate top dead centre conditions of a combustion engine, covering a range of moderate to severe running conditions during ignition. Based on the sliding velocities (stroke and stroke frequencies) and the obtained coefficient of friction (COF), mixed and boundary lubrication conditions are assumed for the tribological experiments. For these test conditions, the relation between wear and the tribofilm structure at the end of the tests with comparable total sliding distance was investigated.

- Coefficient of friction: Higher friction is directly related to higher temperature and lower sliding frequency as expected for the boundary and mixed lubrication regime. The load has no (or hardly any) effect on the friction coefficients measured within this study;
- Wear analysis: The wear results show the load-dependent behaviour that is expected from Archard´s wear model. For the interpretation of the wear trend due to the change in sliding frequency or temperature, the Archard wear model would need an extension taking the tribofilm formation as well as the respective lubrication condition into account;

- Surface analysis: Based on the XPS studies and the comparison with AFM and EDX measurements, a ranking in terms of tribofilm appearance is obtained, which states that higher shear and temperature result in a more pronounced tribofilm. That thickness of the tribofilm is however a matter of debate because of the pad-like geometry of the ZDDP films and therefore more topographic parameters need to be considered in this context.
- Tribofilm and wear: A thicker tribofilm does not automatically imply lower wear or wear rates. According to other research papers in the field, the thicker tribofilm should lead to higher friction but lower wear, which is not necessarily the case based upon the testing conditions in this study. The presented results indicate that a steady-state condition somehow is reached, including tribofilm formation, tribofilm removal and substrate removal. The substrate removal is obvious due to the physical characteristics of the applied wear measurement method as only the substrate has been activated.
- Lubrication regime: The formation of the tribofilm is a consequence of the applied shearing and temperature, and the removal is a consequence of the shearing. As such, the loading conditions need to be considered for better characterising the lubrication regime, which then has to be taken into account for modelling the tribofilm formation and wear behaviour of the tribofilm and substrate.

**Author Contributions:** Conceptualization, M.J.; methodology, M.J.; validation, C.G.; formal analysis, M.J.; investigation, M.L.M.-M. and C.T.; data curation, M.J., M.L.M.-M. and C.T.; writing—original draft preparation, M.J., M.L.M.-M., T.W. and C.G.; writing—review and editing, M.J., M.L.M.-M., T.W., C.T. and C.G.; visualization, M.J., M.L.M.-M. and C.T.; supervision, M.J. and C.G.; project administration, M.J.; funding acquisition, M.J. All authors have read and agreed to the published version of the manuscript.

**Funding:** This research was funded by the Austrian COMET Programme (Project K2 InTribology, Nr. 872176) and has been carried out within the "Austrian Excellence Centre for Tribology" (AC2T research GmbH). The APC was funded by the Austrian COMET Programme (Project K2 InTribology, Nr. 872176).

**Data Availability Statement:** The data presented in this study are available on request from the corresponding author. The data are not publicly available due to privacy restrictions.

**Acknowledgments:** We thank Mitjan Kalin and Sara Spiller for their collaboration during the preparation of the samples and acquisition of AFM measurements.

**Conflicts of Interest:** The authors declare no conflict of interest.

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
