# Peer review of "Effect of Sliding on the Relation of Tribofilm Thickness and Wear"

_lubricants, doi:10.3390/lubricants11020072_

Round 1

Reviewer 1 Report

The research target is good. I recommend this manuscript be published after some minor revisions that will leave it at the level of quality required in this journal. Here are some recommendations:

1-  In the Introduction, the gap of research work (scientific problem) is neither adequately elaborated.

2-  The authors did not show complete characterization for testing samples (piston ring and cylinder liner) such as materials via XRD, primary surface roughness, and hardness.

3-  How many friction and wear tests, please specify in the experimental section.

4-  The amount of lubricant used in any test was not reported.

5-  A protective layer on the worn surface should be confirmed with an EDS line scan on the cross-section of the sample. Otherwise, it is not clear.

6-  SEM and EDS did not well discuss the wear and friction mechanisms. Please, explain the responsible mechanisms for anti-wear clearly.

Author Response

Thank you for your supporting remarks. Our answers and enhancements are summarised in the attached file. 

Reviewer 2 Report

The manuscript is interesting and it could be considered for publication. My suggestions and comments are listed below  

- In the abstracts, it is necessary to more clearly show the novelty of the article.
- The introduction is general, the authors have not mentioned the significance of the present study.
- In the literature review, authors should briefly present some of the existing relevant work emphasizing on the gap between the current work and the existing literature and also should demonstrate how the current work will fill this gap.
- Details on the references and their relative information should be presented. At the current format, combining many references to support one statement is not acceptable.
- You selected a load of 50 and 200 N and a Frequency of 6 and 28 Hz. What was the reason?
- Why did you choose a test duration of 6  and 35 h and a sliding distance of 3240 and 3628 m?
- Specify the material of the contact elements (piston ring and cylinder).
- How many times have the tests been repeated?
 - Can ws in Figure 2 be called steady state wear as there is a significant increase in wear with time?
- T1, T2, T2 ... in fig. 3 - 16 represent the test number, but I can't find the details. Give an exact description for each of these tests.
- "The difference in the loads of 50 N and 200 N does not significantly  influence the value of the friction coefficients. Thus, the effect of load cannot be seen in the coefficients of friction." - Why?
- Line 259: "...trends of the friction coefficients for 20 °C (see Figure 1110)."  - where is Figure 1110?
- Line 290: "...ture leads to an increase of wear (see Figure 1413) but also to an increase of friction (see" - Figure 1413?
- Line 291: "Figure 1614). The increase of temperature goes along with a decrease of viscosity of the " - Figure 1614?
- Line 302: "Regarding wear (see Figure 1315) and friction (see Figure 1516) for tests at 6 Hz (di-" - Figure 1516?
- Line 358: "Table 1... " - should be Table 4.
- Can you indicate the presence of grooves, cracks and delamination on the SEM micrographs?
- Friction and wear mechanisms should be discussed deeply.
- The conclusion should answer the purpose of the study and not simply try to justify the study purpose.

Author Response

(The authors gave the same response as above.)

Round 2

Reviewer 2 Report

The manuscript has been improved and can be published.